# Gaze on the Prize: Shaping Visual Attention with Return-Guided Contrastive Learning

## Abstract

Visual Reinforcement Learning (RL) agents must learn to act based on high-dimensional image data where only a small fraction of the pixels is task-relevant. This forces agents to waste exploration and computational resources on irrelevant features, leading to sample-inefficient and unstable learning. To address this, inspired by human visual foveation, we introduce *Gaze on the Prize*. This framework augments visual RL with a learnable foveal attention mechanism (*Gaze*), guided by a self-supervised signal derived from the agent's experience pursuing higher returns (the *Prize*). Our key insight is that return differences reveal what matters most: If two similar representations produce different outcomes, their distinguishing features are likely task-relevant, and the gaze should focus on them accordingly. This is realized through return-guided contrastive learning that trains the attention to distinguish between the features relevant to success and failure. We group similar visual representations into positives and negatives based on their return differences and use the resulting labels to construct contrastive triplets. These triplets provide the training signal that teaches the attention mechanism to produce distinguishable representations for states associated with different outcomes. Our method achieves up to 2.52× improvement in sample efficiency and can solve challenging tasks that the baseline fails to learn, from the ManiSkill3 benchmark, without modifying the underlying algorithm or hyperparameters.

## 1 Introduction

One of the characteristics of intelligent behavior is the ability to perceive a complex visual world while focusing only on what matters (Itti & Koch, 2001). Through a combination of gaze and foveation, humans naturally discount irrelevant visual information, focusing only on the small subset of cues needed to complete a task (Hayhoe & Ballard, 2005; Land, 2009). For artificial agents (e.g., robots) however, replicating this natural human ability of selective, task-directed attention remains a fundamental challenge. Modern Reinforcement Learning (RL) has achieved great success when provided with structured state representations or carefully engineered features (Mnih et al., 2015; Silver et al., 2016). However, when learning directly from raw pixel inputs, these same algorithms struggle with sample efficiency and robustness. The core issues arise from high-dimensional visual observations that contain vast amounts of task-irrelevant information. Unlike recent studies in supervised learning, where human demonstration can guide where to look, either through active vision (Chuang et al., 2025a; Xiong et al., 2025) or human gaze supervision (Chuang et al., 2025b), RL agents must learn to distinguish relevant and irrelevant features from pixels through their own trial and error. The result is a sample-inefficient process, in which high-dimensional inputs make it difficult to identify the visual cues critical to task success, and cause agents to waste extensive exploration and computational resources.

To address this, we propose *Gaze on the Prize*, a framework that learns visual attention based on simple, yet powerful insights: 1) at any instant of time, task relevant visual cues are localized in one or more contiguous regions within the robot's field of view, 2) when similar states lead to different returns, their distinguishing features are likely to contain task-relevant information (Jonschkowski & Brock, 2015; Blakeman & Mareschal, 2020; Kim et al., 2024). In our approach, the RL agent leverages gaze, a visual attention mechanism to identify task-relevant features, trained via a contrastive signal derived from the agent's reward. For this purpose, we first isolate visual inputs that are similar in the feature space but lead to different reward outcomes. By contrasting such inputs,

the attention mechanism can be guided to discover task-relevant features. Our method is designed to be compatible with any visual RL algorithm and augment their performance. While it adds an auxiliary contrastive objective that affects both attention and policy/value learning, it preserves the core structure and hyperparameters of the base RL algorithms.

Our visual attention provides a lightweight focus mechanism using just five parameters to represent gaze as a Gaussian region, offering human-like inductive bias and also improving the explainability of the agent's actions.

Our contributions can be summarized as:

1. **Learnable Foveal Attention for Visual RL:** We adapt a parametric attention model from human gaze research to visual RL. This design provides an inductive bias that is well-suited for manipulation tasks and provides explainable insights into the agent's actions.

2. **Return-guided Attention Learning**: We devise a contrastive learning method that shapes attention patterns by comparing how an agent's gaze influences the task outcome. Using triplet loss, the attention mechanism learns to focus on image regions that distinguish success from failure.

3. **Plug-in Compatibility:** Our approach serves as a plug-in enhancement that is agnostic to the underlying RL framework. It adds a dedicated gaze module to improve visual RL algorithms while preserving their logic and structure.

4. **Experimental Evaluation:** We demonstrate the utility of the proposed method on RL schemes such as off-policy SAC and on-policy PPO.

## 2 RELATED WORKS

### 2.1 GAZE IN COMPUTER VISION AND ROBOTICS

There is growing interest in leveraging human gaze and gaze-inspired mechanisms in computer vision and robotics. Many works focus on predicting dense saliency maps (Jiang et al., 2015; Kümmerer et al., 2016; 2022; Liu et al., 2020), which identify image regions likely to attract human gaze. Complementary research investigates how saliency maps can enhance downstream performance across a wide range of vision applications (Sugano & Bulling, 2016; Liang et al., 2024; Crum & Czajka, 2025). Other approaches explore parametric attention models that compactly represent gaze and saliency maps via Gaussian or Gaussian mixture models (Song et al., 2024; Reddy et al., 2020).

In robotics, gaze has been studied in both imitation learning (IL) and reinforcement learning (RL). Early IL approaches used Gaussian mixture models to estimate human gaze, foveating a robot's vision by cropping its input around the predicted gaze (Kim et al., 2020). A more recent work leverages foveated vision transformers, allocating high-resolution patches near the gaze and coarser patches toward the periphery (Chuang et al., 2025b). In RL, VisaRL (Liang et al., 2024) pretrains a vision encoder using human-annotated saliency maps, improving downstream performance for robot control. Eye-Robot (Kerr et al., 2025) trains an RL agent to control a mechanical eyeball, with rewards guided by the behavior cloning loss of a co-trained IL policy. In contrast to these approaches, we explore how gaze and attention can emerge naturally within training of standard visual RL frameworks using contrastive learning, without relying on explicit gaze supervision or specialized hardware.

### 2.2 CONTRASTIVE LEARNING IN RL

Contrastive learning has emerged as a powerful approach for representation learning in RL. CURL improves sample efficiency in RL by applying an instance-level contrastive loss to augmented image observations alongside the standard training objective (Laskin et al., 2020). M-CURL extends CURL by adding a masked contrastive objective over video frame sequences, using a transformer to reconstruct features that match the ground truth (Zhu et al., 2022). Eysenbach et al. (2022) reinterprets goal-conditioned RL as a contrastive learning problem, aligning state-action embeddings to approximate a goal-conditioned value function. TACO introduces a temporal contrastive objective to align state-action features with future state representations (Zheng et al., 2023). Also related is

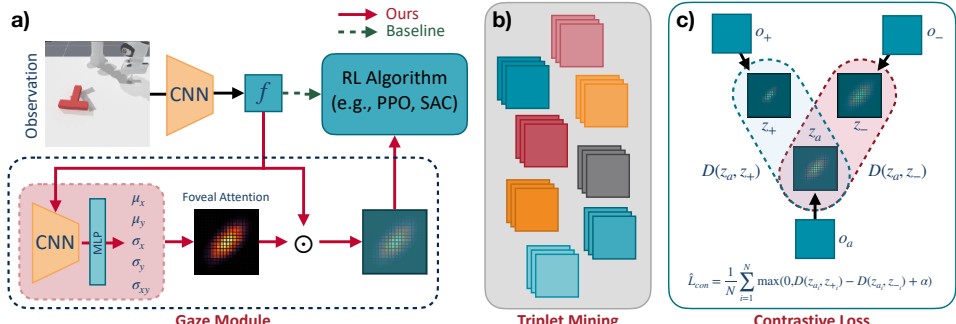

Figure 1: a) A CNN backbone encodes observations into feature maps. Instead of passing them directly to the RL algorithm (baseline), our method refines them with a gaze module that predicts Gaussian attention weights parameterized by $\mu_x, \mu_y, \sigma_x, \sigma_y, \sigma_{xy}$. Multiplying features by these weights ($\odot$) creates a human-like, foveated representation for the RL algorithm. b) During training, we store CNN features and returns in a buffer. Triplet mining groups together similar features that yield different returns. c) The attention is applied on each triplet and a contrastive loss on cosine distances (anchor $z_a$, positive $z_+$, negative $z_-$) guides the module to adjust its attention to better distinguish features by reward.

work by, Liu et al. (2021) which uses reward returns to define a contrastive loss, aligning state-action pairs with similar returns. Our work complements and extends this idea by investigating how return-guided contrastive learning can instead shape visual attention, improving both performance and explainability.

While our objective and mechanism differ from CURL which aims to learn robust visual representations, both operate at the visual representation level and influence how information is extracted from pixel observations. For this reason, we also include a direct empirical comparison to CURL to position our attention learning approach in relation to the broader family of contrastive representation learning methods.

## 3    GAZE ON THE PRIZE

### 3.1    OVERVIEW

The goal of **Gaze on the Prize** is to improve visual RL in manipulation by learning a constrained visual focus to a specific region(s) within a view, similar to how humans focus on task relevant regions while performing tasks. This is achieved through an attention mechanism trained with return-guided contrastive learning. Our framework is designed as a versatile plugin, the main additions being a simple gaze module and an auxiliary contrastive loss that enhances the base RL algorithm. It is fully compatible with any vision-based RL architecture including those with standard CNN-based vision encoders.

The method consists of four components: (1) a learnable foveal attention formulated as a 2D Gaussian function, (2) a contrastive buffer storing CNN features and task returns, (3) a triplet mining procedure to identify similar features that lead to different outcomes, and (4) a return-guided contrastive loss that trains the representation to better distinguish similar features with different returns.

### 3.2    GAZE MODULE

The gaze module is a component that attaches to a standard visual backbone (e.g., NatureCNN (Mnih et al., 2015)) without modifying its architecture (see Figure 1). It generates spatial 2D attention weights by processing the backbone's final feature maps, allowing extraction of weighted information from visual inputs. Inspired by exponential modeling of human foveation (Lehky & Sereno, 2011; Roth et al., 2023; Li et al., 2025), we implement a parametric attention mechanism modeled as an anisotropic Gaussian distribution, which we call *foveal attention*. This introduces a strong inductive bias in RL agent training. The foveal attention mechanism is parameterized by position

$(\mu_x, \mu_y)$ and covariance $(\sigma_x, \sigma_y, \sigma_{xy})$ (see Figure 1). This foveal attention rescales the backbone's feature map element-wise.

This design offers some key advantages: (1) strong inductive bias that we hypothesize aligns with manipulation tasks and constrains the solution space for a more data-efficient learning process, (2) explainable visualizations of the cues driving the agent's actions, and (3) a lightweight module that adds minimal overhead to existing architectures.

### 3.3 CONTRASTIVE ATTENTION LEARNING

Consider a visual observation $o$, and its spatial feature map $f = \text{Enc}(o)$. We assume that $f$ comprises of task-irrelevant features $f_{\text{irr}}$ (e.g., background) and task-relevant features $f_{\text{rel}}$ (e.g., objects of interest). An effective RL policy should be primarily driven by $f_{\text{rel}}$. Hence, *Gaze on the Prize* aims to help the RL agent in more effectively extracting $f_{\text{rel}}$ via an attention mechanism $A_\theta$ with parameters $\theta$. In other words, we aim to find $\theta$ such that $z = f \odot A_\theta(f) \approx f_{\text{rel}}$, where $\odot$ is element-wise multiplication.

The key insight is that when similar visual states lead to different returns, their distinguishing features are likely task-relevant. We formalize this through a contrastive learning framework that shapes attention to highlight these discriminative features.

#### 3.3.1 CONTRASTIVE ATTENTION LOSS

Given an anchor observation $o_a$ with return $R_a$, we identify its $k$-nearest neighbors in the feature space and partition them into two groups based on their returns. Neighbors with returns greater than the anchor's return ($R > R_a$) form the positive pool $\mathcal{P}(o_a)$, while those with lower returns ($R < R_a$) form the negative pool $\mathcal{N}(o_a)$. From each pool, we sample from a predefined top percentile for positives and a predefined bottom percentile for negatives, ensuring that sampled negatives always have lower returns than the anchor and sampled positives always have higher returns. This guarantees a clean ordering in return space and yields stable and well-separated triplets.

For triplets $(o_a, o_+ \in \mathcal{P}(o_a), o_- \in \mathcal{N}(o_a))$, the contrastive loss can be formulated as:

$$\mathcal{L}_{\text{con}}(\theta) = \mathbb{E}_{(o_a, o_+, o_-)} \left[ \max(0, D(z_a, z_+) - D(z_a, z_-) + \alpha) \right] \tag{1}$$

where $D(\cdot, \cdot) = (1 - \text{cosine similarity})$ computed on L2-normalized features, and $\alpha$ is a triplet margin. In practice, we approximate this expectation using batches of $N$ triplets:

$$\hat{\mathcal{L}}_{\text{con}} = \frac{1}{N} \sum_{i=1}^{N} \max(0, D(z_{a_i}, z_{+_i}) - D(z_{a_i}, z_{-_i}) + \alpha) \tag{2}$$

When attention highlights regions that fail to separate anchor-positive from anchor-negative pairs (i.e., when $D(z_a, z_+) \approx D(z_a, z_-)$), the triplet margin $\alpha$ ensures the loss remains positive, generating gradients that push attention parameters toward different spatial locations. Conversely, when attention successfully focuses on discriminative regions ($D(z_a, z_-) - D(z_a, z_+) > \alpha$), the loss reaches zero, stabilizing the current attention parameters. Through optimization over many such triplets, attention converges on features that reliably distinguish different outcome levels.

For Gaussian attention mechanisms like our foveal attention (Section 3.2), we add a regularization term to prevent attention from collapsing to a single point or overly diffusing its coverage:

$$\mathcal{L}_{\text{spread}} = \sum_{i \in \{x, y\}} (\log(\sigma_i) - \log(\sigma_i^{\text{target}}))^2 \tag{3}$$

This log-space formulation regularizes the foveal spread and improves training stability. The complete attention learning objective combines the contrastive loss with regularization:

$$\mathcal{L}_{\text{attn}} = \hat{\mathcal{L}}_{\text{con}} + \lambda_{\text{spread}} \mathcal{L}_{\text{spread}} \tag{4}$$

This attention loss is then integrated with the base RL algorithm's objective:

$$\mathcal{L}_{\text{total}} = \mathcal{L}_{\text{RL}} + \lambda_{\text{attn}}\mathcal{L}_{\text{attn}} \tag{5}$$

where $\lambda_{\text{attn}}$ and $\lambda_{\text{spread}}$ are each hyperparameters controlling the relative importance of attention learning and spread regularization.

### 3.3.2 CONTRASTIVE BUFFER AND RETURN-GUIDED TRIPLET MINING

The contrastive loss (Equation 2) requires triplets of similar features with different returns. We first describe our buffer for storing vision features, then detail how we mine the triplets from this buffer. During training, we maintain a buffer (separate from the on-policy batch or off-policy replay buffer) storing three elements from each observation: (1) detached feature maps from the vision backbone's final layer, (2) flattened feature embeddings for efficient similarity search, and (3) associated episode returns. These features are extracted directly from the same PPO batches or SAC replay-buffer transitions used for RL training and no additional data collection is required. This circular buffer stores diverse historical visual features rather than just recent features. The vision features are stored detached from the computation graph for gradient isolation, preventing the contrastive loss from participating in vision backbone training. During training, stored features pass through the current attention head, creating gradient paths that update attention while preserving the learned vision representations. This design also reduces computational cost and memory requirements compared to backpropagating through the full network.

We implement triplet mining using FAISS (Douze et al., 2024; Johnson et al., 2019) for efficient $k$-nearest neighbor search. FAISS enables sub-linear search complexity even with large search space, making our approach practical for long training runs. A persistent FAISS index is incrementally updated as new samples enter the buffer. During each mining iteration, we sample anchor features from the buffer and retrieve their $k$-nearest neighbors based on cosine similarity of L2-normalized features. These neighbors form candidate pools for triplet construction. We partition them into positive (high-return) and negative (low-return) groups and randomly sample one from each group to pair with the anchor, forming triplets for contrastive learning.

Finally, because our triplet construction relies on distinguishing higher- and lower-return neighbors, the contrastive objective assumes that returns exhibit some degree of variation during training. This requirement is not specific to our formulation but is inherent to return-based contrastive learning in general. For example, RCRL (Liu et al., 2021) constructs positive and negative samples by segmenting trajectories according to their returns, an approach that implicitly relies on return variation to provide informative supervision. However, the return-segmentation used in RCRL assumes that transitions with identical or thresholded returns are behaviorally similar, which is an approximation that becomes unreliable in contact-rich manipulation, where visually and geometrically distinct states often share identical rewards. In contrast, modern manipulation benchmarks such as ManiSkill3 (Tao et al., 2025) and Meta-World (McLean et al., 2025) naturally provide shaped or semidense rewards arising from multi-stage task structure and partial-progress signals, which produce sufficient return diversity for stable triplet construction without requiring additional segmentation and approximation.

## 4 EXPERIMENTS

### 4.1 RESEARCH QUESTIONS

We design our experiments to validate two core contributions: (1) whether parametric foveal attention improves visual RL performance, and (2) whether return-guided contrastive learning enhances this attention mechanism. Specifically, we investigate the following research questions:

**RQ1: How do different attention mechanisms affect visual RL performance?** We compare baseline CNN, patch attention (using 1x1 convolution to generate per-patch attention weights), and our foveal attention on on-policy PPO to understand the impact of attention structure on manipulation tasks.

**RQ2: Does return-guided contrastive learning enhance foveal attention?** We evaluate foveal attention with and without contrastive learning to isolate the contribution of our contrastive learning.

**RQ3: Does return-guided contrastive learning help the agent focus on task-relevant regions in the presence of distractor objects?** We train RL agents in cluttered environments to test whether our attention can learn to filter irrelevant visual information despite distractions.

**RQ4: Is our approach applicable across different RL algorithms?** We validate our complete framework with off-policy SAC to demonstrate compatibility beyond on-policy methods.

**RQ5: How does our approach compare against existing contrastive representation-learning baselines?** We compare our method against CURL, a contrastive learning visual RL method, to evaluate how a standard augmentation-based contrastive objective transfers to visually complex manipulation settings and how it differs from our return-guided attention learning framework.

## 4.2 EXPERIMENTAL SETUP

We evaluate our approach on seven robotic manipulation tasks from the ManiSkill3 benchmark (Tao et al., 2025). The chosen tasks require diverse manipulation skills ranging from simple pushing to object reorientation, providing a comprehensive test of our attention learning approach. For a subset of the tasks, we additionally train and evaluate on variants with random visual clutter to assess whether contrastive attention learns to focus on task-relevant features despite the clutter. Details of each task are provided in Appendix A.

We build on widely used PPO (Schulman et al., 2017) and SAC (Haarnoja et al., 2018) implementations provided in ManiSkill3 and CleanRL (Huang et al., 2022). For baselines, we either use the recommended hyperparameters from the benchmark or lightly tuned variants to ensure stable training. We do not perform heavy task-specific tuning, since our goal is to test whether the foveal attention and return-guided contrastive learning provide consistent improvements to the base RL algorithm *without* altering the base RL training behavior. This ensures that our comparisons reflect the effect of our method rather than differences in baseline optimization. For our foveal attention and return-guided contrastive components, we adopt a single set of default hyperparameters across all tasks: the number of anchors is 1024, the top $k$=16 nearest neighbors are used for sampling positives and negatives, the contrastive buffer holds 100,000 samples, the contrastive loss weight of $\lambda_{attn}$=0.1, and the contrastive objective is applied every iteration. We emphasize that these values were chosen as reasonable defaults and may not be individually optimal for each task, though we include ablation on key hyperparameters to examine their influence on performance. A full hyperparameter table for both RL algorithms and our contrastive components is provided in Appendix D.

## 5 RESULTS

We evaluate our method on a suite of robotic manipulation tasks from ManiSkill3, organized around the research questions in Section 4. Experimental results are averaged over three seeds, with shaded regions in learning curves indicating standard error across runs.

## 5.1 HOW DO DIFFERENT ATTENTION MECHANISMS AFFECT VISUAL RL PERFORMANCE? (RQ1)

We evaluate PPO agents on six ManiSkill3 tasks, comparing three architectural variants: (1) a baseline without attention, (2) patch attention, and (3) our foveal attention, all without contrastive learning. This comparison is to isolate the effect of attention architecture independent of the contrastive learning objective. For the foveal attention without contrastive learning, we retain the same parametric Gaussian attention head but remove the return-guided contrastive supervision. Thus, the attention parameters receive gradients only from the RL objective. This ensures that any performance difference is attributable purely to the architectural structure of the attention mechanism rather than the additional contrastive signal. As shown in the top row of Figure 2, both attention mechanisms increase sample efficiency (measured by steps to 50% success rate) in five out of six tasks compared to the baseline without attention. Additionally, we observe that our foveal attention consistently reaches the 50% success rate faster than patch attention across all tasks. These findings suggest that while spatial attention generally adds value when training visual RL, the architectural choice

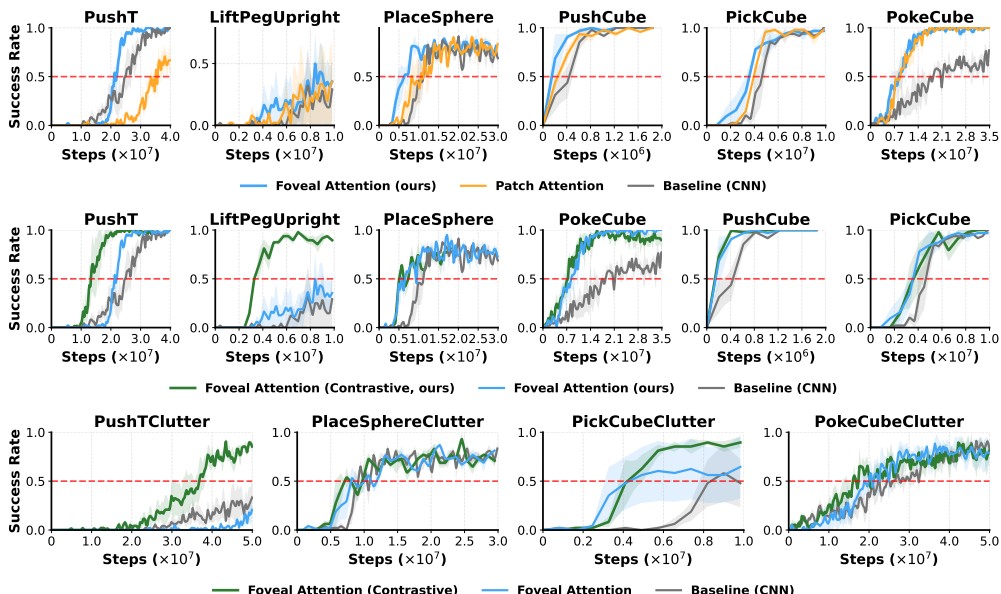

Figure 2: PPO results on ManiSkill3 environments. **Top:** Comparison of attention architectures without contrastive learning (RQ1), showing baseline CNN, patch attention, and foveal attention. **Center:** Effect of return-guided contrastive learning on foveal attention performance (RQ2), comparing foveal attention with and without contrastive supervision. **Bottom:** Comparison of foveal attention with and without contrastive learning when trained in the presence of visual distractor objects (RQ3), demonstrating the robustness benefits of return-guided supervision.

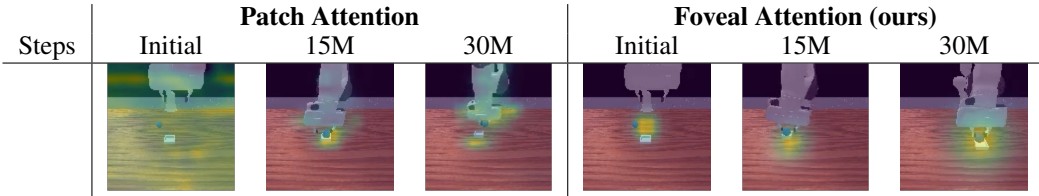

Figure 3: Attention visualization on `PlaceSphere` task across training steps. **Left:** patch attention produces scattered focus across multiple regions. **Right:** foveal attention maintains consistent, concentrated focus. Videos can be found in our supplementary materials, detailed in Appendix E

matters. We observe that structured focus provides a better inductive bias for object-centric manipulation, which usually requires focused localization throughout the task. Interestingly, we noticed that on `PushT` task, patch attention underperforms the baseline, suggesting that unconstrained weights can deteriorate training. Without structural constraints, attention may focus on misleading features or shift too rapidly during training which leads to unstable training. On the other hand, our foveal attention appears to provide essential regularization, preventing these failure modes while maintaining flexibility to focus on task-relevant regions. See Figure 3 for the visualization of the two attention variants on an example task.

## 5.2 Does return-guided contrastive learning enhance foveal attention? (RQ2)

We now add our full method (foveal attention + contrastive learning) to the comparison. The result shown at the center row of Figure 2 exhibits two modes of improvement depending on task difficulty. For simple tasks (e.g., `PushCube` and `PickCube`), the performance benefits of added contrastive learning are marginal. However, for challenging tasks, contrastive learning provides stronger impact, where for `PushT`, contrastive learning provides a $1.86\times$ improvement in sample efficiency to reach

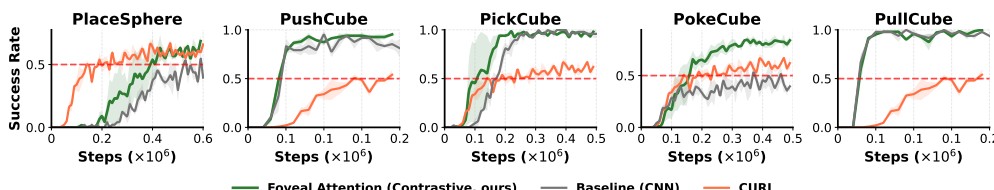

Figure 4: SAC results on ManiSkill3 environments: foveal attention with return-guided contrastive learning, CURL, and the SAC baseline. Across most manipulation tasks, return-guided foveal attention provides the most consistent performance improvements.

50% success, and for `LiftPegUpright`, only the contrastive variant reaches over 50% success rate. Notably, `PokeCube` shows the highest improvement, with 2.52× better sample efficiency compared to the baseline.

### 5.3 DOES RETURN-GUIDED CONTRASTIVE LEARNING HELP THE AGENT FOCUS ON TASK-RELEVANT REGIONS IN THE PRESENCE OF DISTRACTOR OBJECTS? (RQ3)

To further investigate whether contrastive learning helps the attention mechanism to filter irrelevant visual information, we train and evaluate on cluttered variants of a subset of the tasks. As shown in the bottom row of Figure 2, the performance gap is more apparent. For example, while foveal attention without contrastive learning is unable to solve the `PushTClutter` task, even underperforming the baseline, contrastive learning provides the necessary guidance to find critical cues from the cluttered environment. Also for `PokeCubeClutter`, the foveal attention with contrastive learning attains 50% success rate with the fewest environment steps.

Across both clean (RQ2) and cluttered (RQ3) environments, we observe that return-guided contrastive learning provides more value as visual complexity grows. Whether the complexity comes from task difficulty (challenging manipulation requiring fine discrimination) or environmental factors (visual clutter), the contrastive objective helps attention focus on genuinely task-relevant features. This demonstrates that our method enhances performance by learning robust attention patterns that generalize across different sources of visual complexity.

### 5.4 IS OUR APPROACH APPLICABLE ACROSS DIFFERENT RL ALGORITHMS? (RQ4)

To test whether the benefits of return-guided contrastive attention extend beyond on-policy PPO, we evaluate our method with off-policy SAC (Soft-Actor-Critic) (Haarnoja et al., 2018) on five Maniskill3 tasks. As shown in Figure 4, we observe improvements over the baseline, either faster convergence or higher final success rates. The trend is similar to that of PPO, demonstrating that our approach is not tied to a single RL algorithm, but can be applied to different RL methods without heavy modifications.

### 5.5 HOW DOES OUR APPROACH COMPARE AGAINST EXISTING CONTRASTIVE REPRESENTATION LEARNING BASELINES? (RQ5)

We compare our method to CURL (Laskin et al., 2020), a widely used contrastive representation learning approach that has shown strong performance on visually simple domains such as the DM-Control Suite (Tassa et al., 2018). CURL's random-crop augmentation encourages global translation invariance and regularizes the encoder effectively. However, prior work has noted that augmentation-based methods can face challenges such as training instability (Hansen et al., 2021), sensitivity to augmentation types (Yarats et al., 2021), and difficulty in focusing on task-relevant features (Bertoin et al., 2022). We hypothesize that these issues may also occur in manipulation settings, where task-relevant cues are highly localized and depend on fine-grained geometry such as object–gripper alignment or contact geometry.

In our experiments (Figure 4), we observe that CURL performs competitively on certain tasks (e.g., `PlaceSphere`, `PokeCube`), while less effectively on others. We hypothesize that this variation relate to differences in camera configuration, object scale, or scene complexity of the task rather than

any inherent limitation of CURL itself. Across the same tasks, our return-guided foveal attention shows more consistent improvements over the baseline, suggesting that its inductive bias can align well with certain manipulation settings. We emphasize that these observations are empirical and highlight only that different inductive biases may interact differently with the structure and visual requirements of each task. Our implementation of CURL is detailed in Appendix C.

Although both CURL and our approach both use contrastive objectives, they emphasize fundamentally different inductive biases. While CURL promotes global invariance through augmentation-based alignment, our "Gaze on the Prize" objective ties feature similarity to task signals and attentional relevance. These approaches are complementary and the findings suggest exploring hybrid strategies that combine augmentation-based representation learning with reward-guided attention objectives is an interesting direction for future work.

## 5.6 ABLATIONS

We conduct ablations on the `PushT` task under PPO, as this setting shows the largest gap between the foveal attention and our full method with contrastive learning. This makes it the most informative testbed for isolating the effects of each design choice of our method. We focus our ablation on two core hyperparameters that characterize the general contrastive learning pipeline: (1) buffer size, which determines the diversity and temporal coverage of samples available for mining positive and negative pairs, and (2) update frequency, which affects both the computational cost and the consistency of corrective attention signals.

**Buffer size:** Contrastive buffer size is an important design choice of our method, as PPO typically uses only the most recent rollout data unlike off-policy algorithms. Prior work has shown that augmenting on-policy algorithms with replay buffers can improve performance (Wang et al., 2016; Novati & Koumoutsakos, 2019), motivating our use of a separate persistent buffer for contrastive learning. As shown in Fig. 5, using only the on-policy batch ($\approx$20k) still provides improvement over the no-contrastive baseline, demonstrating that even short-horizon features can meaningfully guide

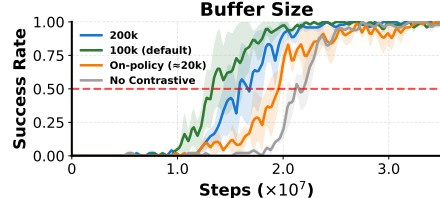

Figure 5: Ablation study on different contrastive buffer sizes.

the foveal attention. Increasing the buffer to 100k yields the best performance, suggesting that a moderate amount of temporal diversity is beneficial for mining informative triplets. Interestingly, a much larger buffer (200k) still outperforms the no-contrastive variant but shows slower early learning, likely due to mining from stale representations that no longer match the current encoder. This highlights a key trade-off where the buffer must be large enough to provide diverse samples, but not to a degree where outdated features can weaken the quality of contrastive supervision.

**Contrastive Frequency** Because contrastive objective adds computational overhead, we evaluate different contrastive update frequencies to check whether reducing the update frequency preserves performance benefits. As shown in Figure 6, performance improves consistently as contrastive updates become more frequent. Updating contrastive learning every iteration achieves the fastest and most stable convergence, while updating every 2 iterations still yields better performance compared to the no-contrastive baseline. In contrast, updating every 4 iterations slows down learning and increases

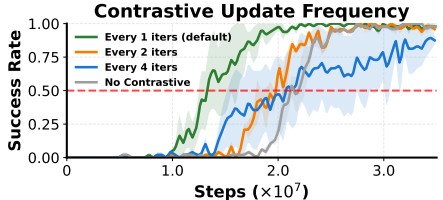

Figure 6: Ablation study on contrastive update frequency.

variance. These results indicate that attention benefits from regular corrective signals and if the contrastive objective is applied less frequently, the learned attention can drift and lose task relevance.

The throughput and performance trade-offs are summarized in Table 1. While contrastive updates reduce throughput (up to 32.8% for every-iteration updates), the resulting gains in sample efficiency outweighs this reduction in throughput. The default setting (contrastive every iteration) achieves a $1.87\times$ improvement in sample efficiency and reaches 50% success in only $0.80\times$ the wall time required by the CNN baseline. Even less frequent contrastive updates (every 2 or 4 updates) still

Table 1: Throughput reduction and performance trade-offs. Steps Per Second (SPS) is measured during training on `PushT` task using a single NVIDIA A6000 GPU, representing wall-clock throughput. Sample efficiency is measured as the ratio of steps required to reach 50% success rate (SR) compared to baseline. Detailed breakdown it provided in Appendix B.

| Config | SPS | Throughput Reduction | Sample Efficiency *(higher is better)* | Wall-time to 50% SR *(lower is better)* |
|---|---|---|---|---|
| Baseline (CNN) | 3820.53 | – | – | – |
| Foveal Attention | 2938.62 | 23.1% | 1.17$\times$ | 1.11$\times$ |
| + Contrastive (every 4) | 2758.58 | 27.8% | 1.19$\times$ | 1.16$\times$ |
| + Contrastive (every 2) | 2669.28 | 30.2% | 1.25$\times$ | 1.15$\times$ |
| **+ Contrastive (every 1)** | **2565.78** | **32.8%** | **1.87$\times$** | **0.80$\times$** |

offer favorable trade-offs, outperforming the baseline in both sample efficiency compared to wall-time gain. These results highlight that the additional compute required for return-guided contrastive learning is not only manageable but in fact yields practical speedups for visual RL. A full breakdown of per-component compute cost is provided in Appendix B.

## 6 LIMITATIONS AND FUTURE WORK

As discussed earlier, our framework assumes that returns exhibit some degree of variation over the course the course of training. This assumption is satisfied under shaped or semi-dense reward settings commonly found in manipulation benchmarks, but becomes weak under strictly sparse-reward environments where most trajectories receive identical or near-identical returns. In such settings, the return-based ordering that drives our triplet construction provides less meaningful signal, leading the contrastive objective to collapse or become uninformative. Recent methods that employ large language models (LLMs) to automatically generate shaped rewards (e.g., Eureka Ma et al. (2023)) offer a promising way to enrich feedback without manual engineering. However, such LLM-driven shaping does not fully resolve the fundamental challenge that sparse environments provide limited task-aligned variation for return-guided contrastive learning. A more direct approach for future work is to incorporate auxiliary signals such as curiosity-driven objectives to augment informative signals even when provided rewards are rare (Burda et al., 2018; Pathak et al., 2017; Badia et al., 2020).

Another structural constraint of our method is that our attention mechanism predicts a single fovea, modeled as a single Gaussian. While this induces a strong and structured inductive bias in certain tasks, some manipulation tasks (e.g., bimanual manipulation) require simultaneously attending to multiple spatially distinct regions within an image. An extension of our framework could predict multiple Gaussian heads (i.e., a mixture of Gaussians) with learned combination weights and adapt the same return-guided contrastive objective to supervise multiple focused regions at once. This multi-fovea variant may better capture richer spatial structure and handle tasks where multiple regions within the visual input carry task relevant information.

## 7 CONCLUSION

In this work, we introduced ***Gaze on the Prize***, a framework that guides attention to focus on task-relevant visual features in RL through return-guided contrastive learning. By contrasting similar states with different outcomes, our method guides attention toward the features that matter for task success. Experiments across multiple robotic manipulation tasks and validation on two RL paradigms (on-policy and off-policy RL) show that our method can enhance standard visual RL with learned attention, improving sample efficiency and performance without requiring architectural changes to the base RL algorithms. To enable visual RL agents to tackle complex, cluttered manipulation tasks, we need methods that can discover task-relevant visual cues without human supervision. We believe this work is a step toward more sample-efficient visual RL, where agents learn not only what to do, but also where to look and focus, similar to the gaze mechanism that makes human vision so effective.

## 8 REPRODUCIBILITY STATEMENT

We designed our experiments to be fully reproducible. Our implementation builds on publicly available PPO and SAC codebases, and hyperparameters are either taken from benchmark defaults or fixed globally across tasks to avoid task-specific tuning. We fix random seeds, report results over three independent runs, and log all training curves and evaluation metrics in a standardized format. Upon acceptance, we will release the complete codebase, including training scripts, modified environments, and hyperparameter tables, together with detailed instructions to reproduce the results in the paper.

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

# A TASK/ENVIRONMENT DETAILS

In our experiments, we evaluate seven manipulation tasks from the ManiSkill3 benchmark (Tao et al., 2025). All tasks use single-camera RGB observations and are solvable by PPO within reasonable training time, ensuring fair comparison across methods.

## A.1 TASK DESCRIPTIONS

**PickCube:** Pick up a cube and lift it to a target (target provided as state). Tests basic grasping and lifting.

**PushCube:** Push a cube to a target location on the table surface. Tests object manipulation without grasping.

**PullCube:** Pull a cube to a target location on the table surface. Tests object manipulation without grasping.

**PokeCube:** Poke a cube with a peg to a target location on the table surface. Tests tool use and indirect manipulation.

**PushT:** Push a T-shaped object to match a target pose. Tests manipulation requiring rotation and translation.

**LiftPegUpright:** Reorient a peg from horizontal to an upright position. Tests complex reorientation.

**PlaceSphere:** Pick up a sphere and place it in a slot. Tests precise placement and object handling.

## A.2 OBSERVATION AND ACTION SPACES

All tasks share consistent observation and action specifications:

- **Observation:** 128×128 RGB images (64x64 for SAC) from a fixed camera position
- **Action:** 7-DOF continuous control
- **Episode length:** 50-100 steps depending on task

## A.3 REWARD STRUCTURE

Tasks use dense reward signals combining distance-based rewards for approaching intermediate targets, and success bonuses for achieving goals. This dense reward structure enables meaningful return differences for contrastive learning.

## A.4 CLUTTERED ENVIRONMENTS

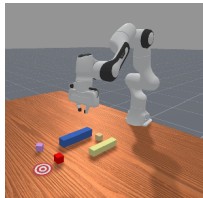 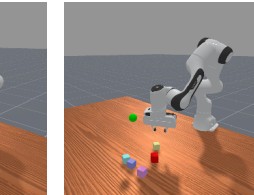 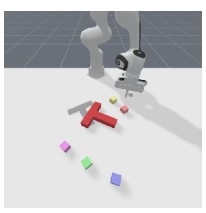 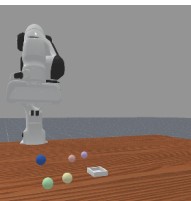

| PokeCubeClutter | PickCubeClutter | PushTClutter | PlaceSphereClutter |

Figure A1: Visualization of cluttered environments for robustness evaluation. Visual distractors include objects similar in shape and color to task-relevant items.

To evaluate whether return-guided contrastive learning help the agent focus on task-relevant regions in the presence of distractor objects (RQ3, Section 5.3), we create cluttered variants of four tasks by adding randomly placed distractor objects to the scene. These cluttered environments are designed to challenge the agent's visual processing by introducing objects that are visually similar to task-relevant items but functionally irrelevant.

# B  COMPUTE OVERHEAD ANALYSIS

To better understand the computational cost of our proposed components, we compare the baseline CNN encoder, the foveal-attention variant without contrastive learning, and the full method with both foveal attention and return-guided contrastive learning. All experiments run with 1024 parallel environments, 16-step rollouts, and a minibatch size of 512, using the same single NVIDIA RTX A6000 GPU. This controlled setup ensures a fair and reproducible comparison of compute overhead across variants.

Table B1: **Compute and overhead comparison across model variants.** Values are reported in milliseconds per update unless noted otherwise. Overheads for foveal attention and contrastive learning are computed relative to the total update time for each model.

| Component | Baseline CNN | Foveal (no contrast) | Full Method |
|---|---|---|---|
| **Training Throughput** | | | |
| Steps per Second (SPS) | 3820.53 | 2938.62 | 2565.78 |
| Relative SPS | 100% | 76.9% (-23.1%) | 67.2% (-32.8%) |
| Wall Time to 1M steps (seconds) | 261.59 | 340.10 | 389.52 |
| **Update Cost (ms)** | | | |
| Total Update Time | 2091.58 | 3247.96 | 3841.16 |
| Foveal Attention Forward | — | 1.30 | 1.387 |
| Contrastive: Triplet Mining | — | — | 487.44 |
| Contrastive: FAISS Search | — | — | 4.09 |
| Contrastive: Buffer Add | — | — | 0.15 |
| **Relative Overhead (%)** | | | |
| Foveal Attention Overhead | — | (1.30 / 3247.96 = 0.04%) | 0.04% |
| Contrastive Learning Overhead | — | — | 12.80% |

Across all variants, the majority of compute lies in the standard PPO update (forward passes through the CNN encoder, policy, and critic, along with backpropagation). Foveal attention introduces a negligible forward-pass cost ($<0.05\%$), while the majority of contrastive overhead arises from triplet mining, which itself contributes about 12–13% relative to the full update cost.

All per-component timings are measured using standard wall-clock timing with asynchronous GPU execution. As a result, the absolute millisecond values should be interpreted as approximate rather than precise hardware benchmarks. However, because all variants are profiled under identical conditions, the relative overhead percentages (e.g., foveal attention vs. baseline CNN vs. contrastive components) remain directly comparable and are the primary metric of interest in our compute analysis.

# C  CURL IMPLEMENTATION DETAILS

We implement a full CURL baseline following Laskin et al. (2020), while replacing the original convolutional encoder with the same backbone used in our SAC agents to ensure strict apples-to-apples comparison. The agent uses $64 \times 64$ RGB observations and a convolutional encoder that outputs a 256-dimensional feature shared across the actor and critic. CURL adds a bilinear contrastive head and a separate momentum encoder updated via EMA ($\tau_{\text{curl}}$=0.005). For each sampled batch, two independently augmented views are generated using vectorized random shifts with 4-pixel padding. The main encoder processes the query view, while the momentum encoder processes the key view, and the InfoNCE loss is computed from the resulting $B \times B$ similarity matrix. The CURL auxiliary loss is added to the critic update with weight $\lambda_{\text{curl}}$=0.1, while the actor uses encoder features detached from gradient flow. The replay buffer, training pipeline, and all SAC hyperparameters remain unchanged, ensuring that CURL differs from our main method only in the addition of the contrastive objective and momentum encoder.

## D  HYPERPARAMETERS

Table C1: Hyperparameters used in experiments. All values are fixed for main experiments across tasks unless marked with *. Most hyperparameters are adopted from the original baseline implementations (Tao et al., 2025; Huang et al., 2022) with adjustments for attention and contrastive learning components.

| HYPERPARAMETER | VALUE | DESCRIPTION |
|---|---|---|
| *Gaze Module* | | |
| Architecture | [64→32→5] | Conv→ReLU→MaxPool→Linear |
| $\sigma_{\text{target}}$ | 0.1 | Target spread in normalized coords |
| $\lambda_{\text{spread}}$ | 0.1 | Spread regularization weight |
| | | |
| *Contrastive Learning* | | |
| $k$-neighbors | 16 | Neighborhood size |
| Triplet margin ($\alpha$) | 0.5 | Minimum separation distance |
| Contrastive buffer size | 100k | FAISS-indexed buffer |
| Update frequency | 1 | Every N iterations |
| Anchor samples | 1024 | Per triplet mining |
| Top percentile | 50 | Sampled from high-return pool |
| Bottom percentile | 50 | Sampled from low-return pool |
| $\lambda_{\text{attn}}$ | 0.1 | Contrastive loss weight |
| | | |
| *RL Training (PPO)* | | |
| Learning rate | 3e-4 | Adam optimizer |
| Discount factor ($\gamma$)* | 0.8-0.99 | Task-specific |
| GAE $\lambda$ | 0.9 | Advantage estimation |
| Num environments* | 1024-2048 | Parallel environments |
| Num steps* | 4-16 | Rollout length per environment |
| PPO clip range | 0.2 | Policy update constraint |
| PPO epochs | 8 | Updates per rollout |
| Minibatches | 32 | Gradient updates per epoch |
| Target KL | 0.2 | Early stopping threshold |
| Input resolution | 128×128 | RGB images |
| Control mode | pd_joint_delta_pos | Joint-space control |
| | | |
| *RL Training (SAC)* | | |
| Learning rate | 3e-4 | Adam optimizer |
| Discount factor ($\gamma$)* | 0.8-0.99 | Task-specific |
| Num environments | 32 | Parallel environments |
| Batch size | 512 | Samples per update |
| Replay buffer size | 300k | Experience storage |
| Update-to-data (utd) | 0.5 | Gradient steps per env step |
| Target update rate ($\tau$) | 0.01 | Soft update coefficient |
| Input resolution | 64×64 | RGB images |
| Control mode | pd_ee_delta_pos | End-effector control |

## E  ADDITIONAL SUPPLEMENTARY MATERIALS

We provide attention visualization videos and source code as supplementary materials with this submission. For review purposes, these materials can be accessed in the `videos/` and `code/` directory of the supplementary materials. For convenience, an anonymized `index.html` file is provided for browsing of all videos. Upon acceptance, all videos and the complete codebase will be made publicly available at a project website.

