# OpenReview forum: "Gaze on the Prize: Shaping Visual Attention with Return-Guided Contrastive Learning"
_ICLR.cc/2026/Conference — Submitted to ICLR 2026_

### Official Review · Reviewer_eqoC · 2025-10-28

**Soundness:** 2
**Presentation:** 3
**Contribution:** 2
**Rating:** 4
**Confidence:** 3

**Summary:**

The paper introduces a framework for visual reinforcement learning that trains a foveal attention mechanism using triplet data, where two similar feature representations correspond to different returns. The foveal attention module is optimized through return-guided contrastive learning.

**Strengths:**

1. The paper is engaging, presenting an intuitive yet effective idea that is both interesting and easy to follow.

2. The authors conduct extensive experiments, demonstrating the strong performance of the proposed framework and its high flexibility across different algorithms.

**Weaknesses:**

The authors may benefit from providing additional implementation details and a more thorough discussion of the framework’s limitations.

1. Clarification on implementation and data mining: I would appreciate a more detailed explanation of Sections 3.3.2 and 4.1–4.2, where the authors discuss return-guided triplet mining and the model implementation. The proposed method appears to rely heavily on datasets that provide dense reward signals and a large number of feature pairs with similar representations but different returns, which may not be readily available in many practical settings. More details on the data-mining process, dataset characteristics, and implementation specifics would help readers better understand the method’s applicability.
Additionally, the experiments include a baseline model (“Foveal attention” without contrastive loss). It would be helpful to elaborate on this model’s setup to clarify the standalone impact of the contrastive loss component.

2. Extension of the gaze module: The current gaze module focuses on a single focal point, which may limit its generalizability. The authors are encouraged to discuss whether the framework could be extended to handle multiple focal locations. If such an extension is feasible, it would be useful to describe the required adaptations; if not, clarifying the bottlenecks or constraints preventing this would strengthen the discussion.

3. Data availability and robustness: A key limitation, also noted by the authors, concerns the availability of return signals or, more broadly, the scarcity of triplet-mining data. It would be valuable to provide insights into how much effective data is required for training, and how data scarcity might influence performance or stability. This discussion would help readers understand the framework’s robustness under less ideal data conditions.

**Questions:**

see comments above.

---

> ### Author Response · Authors · 2025-11-21
> **Official Response to Reviewer eqoC**
>
> We thank the reviewer for the thoughtful and constructive feedback. We appreciate your detailed reading of our work and address your comments below.
>
> 1. Clarification on implementation and return-guided triplet mining
>
> Our triplet mining procedure operates directly on the standard replay buffers used in SAC and PPO batch, and does not require any additional datasets or data collection. During training, each anchor frame retrieves its top-k nearest neighbors in feature space from a persistent FAISS index that is built from replay-buffer features. These neighbors are then divided into higher-return and lower-return groups relative to the anchor’s return. Positives are sampled from the highest-return portion of the higher-return group, and negatives from the lowest-return portion of the lower-return group. This allows the method to rely entirely on environment-provided episode returns.
>
> In the revised manuscript, we will expand Sections 3.3.2 and 4.1–4.2 to clarify how the FAISS index is maintained, how episode returns are computed and stored, and how triplets are mined online at each gradient step.
>
> Regarding the “Foveal Attention (no contrastive loss)” ablation, this model uses the same Gaussian attention head but removes the return-guided loss. The attention parameters therefore receive gradients only from the RL objective. This isolates the role of the contrastive supervision, since without return guidance the learned attention tends to drift or collapse, while with guidance it forms consistent and structured foveation. We will clarify this setup more explicitly.
>
> 2. Extension of the gaze module to multiple focal points
>
> Thank you for raising this interesting direction. Our current module predicts a single anisotropic Gaussian because many manipulation sub-tasks are naturally dominated by one key visual region at a time. This choice also keeps computation lightweight and preserves interpretability. Empirically, we find that most ManiSkill3 tasks only require attention to one dominant region at any given stage.
>
> However, the framework can be extended to multiple focal points. A possible extension would involve predicting multiple Gaussian heads (i.e., a mixture of Gaussians), combining them with learned weights, and adapting the triplet mining process to operate over multiple foveal regions. While feasible, this introduces additional parameters and computation, and would require designing appropriate multi-region credit assignment for return-guided supervision.
>
> We believe such an extension may be especially useful for bimanual or multi-object manipulation tasks where multiple spatial regions matter simultaneously.  We view this as a promising direction, but one that requires first identifying tasks where multiple attention locations are truly necessary and carefully designing the corresponding multi-fovea supervision. We plan to include a short discussion of this future direction in the revised manuscript.
>
> 3. Data availability, required amount of triplet-mining data, and robustness under scarcity
>
> We agree that diverse returns are important for effective triplet mining. In shaped or semi-dense reward settings used in common manipulation benchmarks such as ManiSkill3, Meta-World, and RLBench, trajectories naturally produce a range of returns, which supports stable positive and negative sampling. When the return distribution collapses, for example when most episodes have return zero, triplet mining becomes ineffective. Increasing the top-k retrieval window is possible but becomes computationally expensive and does not resolve the fundamental issue of lacking reward variation.
>
> Our method therefore assumes that the environment provides moderate return diversity over the course of training. This assumption holds for most modern manipulation domains, and we will clarify it explicitly in the revised version. We acknowledge that strictly sparse-reward environments constitute a genuine limitation for return-based contrastive learning. Designing attention supervision that remains informative under extreme return scarcity is an interesting direction for future work.
>
> We hope these clarifications address the reviewer’s questions. Please let us know if any additional details would be helpful, and we will be happy to provide further clarification.

---

> > ### Comment · Area_Chair_CEsH · 2025-11-22
> >
> > Hi Reviewer,
> >
> > The authors have submitted their responses to your reviews. Please take a look and let the authors know if you have any further questions or concerns. Thank you again for your contributions to ICLR!
> >
> > Best regards,
> > AC

---

> > ### Author Response · Authors · 2025-11-27
> > **Follow-up Response to Reviewer eqoC**
> >
> > We thank the reviewer again for the constructive feedback on our work. As mentioned in our previous response, we have updated the manuscript to address the raised concerns:
> >
> > 1. Clarification on implementation and return-guided triplet mining
> > 3. Data availability, required amount of triplet-mining data, and robustness under scarcity
> >
> > We have revised Sections 3.3.2–4.2 to provide a clearer description of our implementation, including how the contrastive buffer is maintained, how episode returns are stored, and how online triplet mining is performed using FAISS. We also explicitly state the assumptions required for reliable triplet construction and discuss robustness under data scarcity.
> >
> > We ensured that the setup of this ablation model is clearly described in Section 5.1, highlighting that it uses the same Gaussian attention head but without return-guided supervision, allowing us to isolate the architectural effect of foveal attention.
> >
> > 2.	Extension of the gaze module to multiple focal points
> >
> > As suggested, extending the gaze module to multiple foveal regions is an interesting future direction. We added a discussion in Section 6 (Limitations and Future Work) outlining how multi-foveal attention could be implemented.
> >
> > We appreciate the reviewer’s insights and believe these additions help clarify the method and further contextualize its potential extensions.

---

### Official Review · Reviewer_6vmd · 2025-10-30

**Soundness:** 3
**Presentation:** 3
**Contribution:** 2
**Rating:** 4
**Confidence:** 4

**Summary:**

The paper proposes an approach to learn a foveated attention mechanism where attention is encouraged to focus on task-relevant information. To achieve this, the authors propose using a Gaussian parametric attention mechanism paired with a contrastive loss where positive and negative samples are mined based on visual similarity and return differences. For that the authors use FAISS to store tuples of observations and returns, and then for each anchor select the positives and negatives based on their corresponding return classification based on a threshold. In simulated experiments, the approach is shown to improve sample efficiency and performance over model-free on-policy and off-policy algoriths such as PPO and SAC.

**Strengths:**

- The paper is well-written and enjoyable to read
- The proposed approach makes a lot of sense and is sound
- The proposed approach is very simple and can be easily plugged into most vision-based RL frameworks
- The results are promising, the proposed approach seems to yield a consistent improvement over SAC and PPO without strongly impacting wall-clock time, which could have been a concern given the use of retrieval methods such as FAISS
- Ablations illustrate the role of different components and the sensitivity to different hyperparameters that were introduced in this work

**Weaknesses:**

- On the method' side, one detail does not make full sense to me. I do not understand why the positives are chosen to have the higher returns, the anchor might belong to the low-return set and in that case the latter should be treated as positives.
- The main weakness of this work lies in its evaluation. Specifically, the baselines considered in this work fail to position the contribution well. Let me elaborate. While the proposed framework seems to be quite helpfull when using vanilla PPO or SAC, it would be interesting to see if it has any effect on stronger visual RL methods such as CURL or more recent methods. Without such a baseline, it is unclear whether the proposed framework is even needed. In theory, the proposed method should help no matter what baseline is used, but this needs to be validated and well-studied to understand what level of task complexity is needed for this to become apparent.
- Another missing experiment is to understand the role of this plugin  if the encoders are pretrained either large-scale on images or in a first stage with random data from the target environment
- (optional) what would strengthen the paper a lot and increase its impact is to also test the method on a model-based RL baseline, but I understand this can be time-expensive

**Questions:**

- Can you explain the choice of positive and negative sampling in your contrastive learning setup (refer to my first comment under weaknesses)?
- how would your method perform when augmenting stronger vision-based RL baselines such as CURL?

---

> ### Author Response · Authors · 2025-11-21
> **Official Response to Reviewer 6vmd**
>
> We thank the reviewer for the thoughtful and constructive review. We address your comments below.
>
> 1. Regarding the positive and negative sampling
>
> As correctly noted, the previous median-based split did not guarantee a consistent ordering (negative < anchor < positive), leading to ambiguous supervision. We have replaced this with an anchor-centered strategy: after retrieving the top-k neighbors, we (i) form the positive group as neighbors with higher return than the anchor, (ii) form the negative group as neighbors with lower return, and (iii) sample positives from the highest-return percentile and negatives from the lowest percentile. This guarantees clean ordering and avoids contradictory labels.
>
> We are re-running all experiments with this corrected sampling. Early results show reduced variance and improved consistency across tasks. The final updated curves will be posted as they become available.
>
>
> 2. Regarding stronger visual RL baselines such as CURL
>
> We agree this is an important comparison. We are currently running CURL+SAC on ManiSkill3 tasks; preliminary results show instability on contact-rich 3D manipulation, consistent with prior findings, but we will report full curves once the runs finish. We are also implementing our method on DMControl and DMControl-Generalization Benchmarks to enable direct comparisons in CURL’s native domain.
>
> Conceptually, augmentation-based methods (RAD/DrQ-v2/SVEA/CURL) aim to enforce invariance to pixel-level perturbations, while our method targets spatial credit assignment. Because they operate on different layers of the visual stack, they are complementary rather than substitutions. Combining both is deemed possible but non-trivial. CURL’s heavy augmentations can distort the fine-grained spatial cues required for manipulation, and both methods introduce separate contrastive objectives that may interfere during joint optimization. For these reasons, we consider the combination out of scope for the current submission but plan to explore it in follow-up work.
>
>
> 3. Regarding pretrained encoders
>
> Large-scale pretrained encoders (ImageNet, MAE) are rarely used in manipulation RL because natural-image invariances often conflict with the fine-grained geometric cues needed for precise 3D control. Pretraining on target-domain random data is more reasonable and has shown mixed but sometimes positive results. Our attention framework is compatible with such initializations, but evaluating all associated design choices (pretraining loss, stability, architecture variants) is beyond the scope of this submission.
>
> Our focus here is specifically on policy-aligned attention learning, independent of encoder initialization. We plan to explore domain-pretrained encoders as future work.
>
> We hope these clarifications address the reviewer’s concerns and better highlight the scope, contributions, and limitations of our work. As mentioned, we will post the updated experimental results and additional comparisons as soon as the ongoing runs complete. Please let us know if any further clarification would be helpful.

---

> > ### Comment · Area_Chair_CEsH · 2025-11-22
> >
> > Hi Reviewer,
> >
> > The authors have submitted their responses to your reviews. Please take a look and let the authors know if you have any further questions or concerns. Thank you again for your contributions to ICLR!
> >
> > Best regards,
> > AC

---

> > ### Author Response · Authors · 2025-11-27
> > **Follow-up Response to Reviewer 6vmd**
> >
> > We thank the reviewer again for the thoughtful suggestions that helped strengthen our work. As noted in our previous response, we have updated the manuscript to incorporate the new results and clarifications requested by the reviewer.
> >
> > 1. Positive and negative sampling for triplet mining
> >
> > Section 3.3.1 has been revised to reflect the improved return-guided triplet mining procedure.
> > We now explicitly ensure a consistent ordering of
> > $R_- < R_a < R_+$
> > when constructing triplets, resolving the ambiguity present in the original description.
> >
> > 2. Comparison against stronger visual RL baselines such as CURL
> >
> > We added a new research question (RQ5) dedicated to evaluating our approach against existing contrastive-based representation learning methods (Section 4, Section 5.5).
> >
> > Following the reviewer’s suggestion, we now include CURL as a baseline in our SAC experiments.
> > Figure 4 and Section 5.5 provide a detailed comparison and discussion of CURL’s performance relative to ours across multiple ManiSkill tasks.
> >
> > We again appreciate the reviewer’s time and constructive feedback, and we are happy to address any further questions.

---

### Official Review · Reviewer_LYqW · 2025-10-31

**Soundness:** 3
**Presentation:** 3
**Contribution:** 3
**Rating:** 6
**Confidence:** 4

**Summary:**

The paper proposes Gaze on the Prize, a visual reinforcement learning framework that enhances sample efficiency and stability by integrating a learnable foveal attention mechanism inspired by human visual focus. Instead of processing all image features equally, the method guides attention toward task-relevant regions using a self-supervised signal derived from return differences—comparing states that lead to high versus low rewards. Through return-guided contrastive learning, the system learns to emphasize visual features that distinguish success from failure, forming contrastive triplets based on outcome differences. Without changing the base RL algorithms or hyperparameters, this approach yields up to 2.4× higher sample efficiency and enables agents to solve manipulation tasks in the ManiSkill3 benchmark that standard methods fail to learn.

**Strengths:**

1. The paper introduces a biologically inspired, learnable foveal attention mechanism that enhances both the interpretability and efficiency of visual reinforcement learning.
2. The paper provides a flexible, plug-and-play framework that can seamlessly integrate with various RL algorithms to consistently improve performance across manipulation tasks.
3. The paper is well-written, presenting a straight-forward idea with conceptual clarity and a logical narrative that is easy to follow.

**Weaknesses:**

1. One potential drawback of the proposed method lies in that it assumes that there are a sufficient number of clearly distinguishable positive and negative samples. However, in some difficult or sparse-reward tasks, there can be very few positive samples in the replay buffer, which may affect the performance of the policy.

**Questions:**

1. I'm curious about why the episode return instead of step reward is used to guide the contrastive learning, since one-step reward seems a better indicator for the one-step observation.
2. What will happen if the task involves two objects? For example, if the task is to pick a cube into a box, will the attention mechanism always capture both cube and box, or first focus on cube and then box?

---

> ### Author Response · Authors · 2025-11-21
> **Official Response to Reviewer LYqW**
>
> We thank the reviewer for the thoughtful and positive evaluation of our work. We appreciate your clear understanding of our contributions and your constructive feedback. We address your comments and questions below.
>
>
> 1. Regarding the weakness on sparse positive samples
>
> We agree that sparse-reward environments can yield very few successful trajectories, reducing the availability of informative positive samples. When most returns are zero, the return distribution collapses, and with top-k retrieval, neighbors will not exhibit meaningful variation, naturally weakening the contrastive signal. While increasing top-k is possible, it becomes computationally expensive even with FAISS and does not fundamentally resolve the lack of return diversity. Thus, strictly sparse-reward settings represent an inherent limitation for return-based contrastive learning.
>
> In this paper, we focus on shaped or semi-dense rewards (as provided by most modern manipulation benchmarks), and we will clarify this assumption explicitly. We view extending the method to partially or fully sparse-reward domains, potentially including Atari as an interesting direction for future work.
>
> 2. Why we use episode return instead of step reward
>
> Although a one-step reward may seem more tightly coupled to a single observation, in practice it provides a very weak and often misleading training signal. We use episode return instead for the following reasons:
>
> i) Step rewards do not reliably reflect task progress. From the perspective of an individual frame, high step-wise rewards may arise from shaping terms or reward-engineering artifacts, and not necessarily indicate that the agent is closer to solving the task or exhibiting genuine successful behavior.
>
> ii) Step rewards offer poor discriminative power for contrastive learning.
> They are frequently constant or noisy across time, making it difficult to distinguish meaningful differences between observations. This leads to weak or unstable contrastive labels.
>
> iii) Episode return provides a stable, outcome-aligned supervision signal.
> It summarizes the overall quality of a trajectory, enabling consistent ranking of high-performing versus low-performing behaviors and naturally addressing credit-assignment issues that per-step rewards cannot resolve.
>
> We note that although the supervision label is derived at the episode level, the contrastive loss itself operates at the frame level. Each frame inherits the return label of its originating trajectory, enabling robust frame-wise representation learning without requiring finely designed or informative step-wise rewards.
>
>
> 3. What happens when the task involves multiple objects
>
> Many ManiSkill3 tasks (evaluated in this paper) already involve multiple interacting objects, such as PokeCube (the gripper picks up a peg and then uses it to poke a cube) and PlaceSphere (the gripper grasps a sphere and places it into a container). In these settings, our foveal attention mechanism naturally learns to shift its focus as the task progresses, attending to whichever object or region is most predictive of long-term success.
>
> This behavior emerges because the contrastive supervision is return-guided where network learns to emphasize the visual cues that consistently correlate with successful rollouts.
> This also relates to shaped rewards commonly used in manipulation benchmarks. Shaping terms  (e.g., reaching object A, then object B) implicitly encode the temporal structure of a multistage task. For example, reaching the peg first, then interacting with the cube on PokeCube task. Since these shaped signals accumulate into the episode return, frames corresponding to intermediate subgoals naturally receive slightly higher returns, enabling the attention to transition between relevant objects as needed.
>
> Overall, the model simply learns, through return-based contrastive learning, which parts of the scene best distinguish high-performing trajectories from low-performing ones.
>
> We hope these clarifications address the reviewer’s questions and concerns. Please let us know if further details would be helpful.

---

> > ### Comment · Area_Chair_CEsH · 2025-11-22
> >
> > Hi Reviewer,
> >
> > The authors have submitted their responses to your reviews. Please take a look and let the authors know if you have any further questions or concerns. Thank you again for your contributions to ICLR!
> >
> > Best regards,
> > AC

---

> > ### Comment · Reviewer_LYqW · 2025-11-27
> >
> > Thank you for your replies. Although most of my concerns are resolved, the proposed method is limited to shaped or semi-dense reward settings. Meanwhile, as other reviewers have mentioned, comparisons with SOTA visual RL baselines are also expected in the experiment part. Therefore, I will temporarily keep my overall score unchanged.

---

> ### Author Response · Authors · 2025-11-27
> **Follow-up Response to Reviewer LYqW**
>
> Thank you for the follow-up and for taking the time to revisit our responses.
>
> We would like to note that the updated manuscript (uploaded after your recent comment) now explicitly states the shaped/semi-dense reward assumption and includes new comparisons with CURL (Laskin et al.), a widely used SOTA contrastive visual RL baseline. These results are reported in Section 5.5 and Figure 4. Although CURL targets a different aspect of visual RL (global invariances through augmentation) than our reward-guided attention learning, it represents the most relevant contrastive-learning baseline for a direct comparison, and we provide a detailed comparison under manipulation setting.
>
> We also included our CURL implementation details under Appendix C.
>
> We sincerely appreciate your continued feedback and are happy to address any further questions.

---

### Official Review · Reviewer_iVWx · 2025-10-31

**Soundness:** 2
**Presentation:** 4
**Contribution:** 2
**Rating:** 2
**Confidence:** 4

**Summary:**

The paper introduces a lightweight "gaze" head that predicts an anisotropic Gaussian to reweight spatial CNN features, trained with a return-guided triplet loss mined from a separate buffer of detached features, and combined with a standard RL loss (PPO/SAC). On ManiSkill3 manipulation tasks, the method learns faster and shows larger gains under cluttered visuals.

**Strengths:**

- Simple, interpretable, drop-in attention module with few parameters and human-readable heatmaps.

- Clear optimization split: contrastive loss updates the gaze head; RL gradients train the encoder/policy.

- Works with both off-policy algorithm and off-policy algorithm

- Useful ablations (e.g., auxiliary buffer, update frequency) and some wall-clock considerations.

**Weaknesses:**

1. Missing strong pixel-RL baselines. The paper omits standard data-augmentation and contrastive baselines (RAD, DrQ-v2, SVEA, CURL) that are now expected in visual control comparisons. Without these, it is unclear whether the gains stem from the proposed attention or simply from adding any robust representation trick.
2. Insufficient OOD evaluation. Claims about distraction robustness/generalization are not tested on canonical benchmarks with held-out domains and graded shifts (DCS; DMC-GB/GB2). Results limited to ManiSkill3 and custom clutter do not substitute for these protocols.

3. Sensitivity to reward structure. The return-guided triplet objective relies on reward spread; sparse-reward regimes may weaken positives/negatives. A value- or TRP-based auxiliary could help, but this is not explored. ([ojs.aaai.org][3])
5. The narrative hints at generalization, but the experiments emphasize sample-efficiency on manipulation. Without in-distribution to OOD gap reporting, distraction-intensity curves, or cross-background transfer, the generalization claim is under-supported.
6. Related-work positioning. Attention/masking approaches targeting distraction robustness: MaDi; saliency-guided methods like SGQN and SGFD; segmentation-assisted FTD are not compared and deserve deeper discussion. ([ifaamas.org][4])


References:
* Laskin, M., et al. (2020). Reinforcement Learning with Augmented Data (RAD). NeurIPS.[1]
* Stone, A., Ramirez, O., & Jonschkowski, R. (2021). The Distracting Control Suite. [2]
* Wang, S., Wu, Z., Hu, X., Wang, J., Lin, Y., & Lv, K. (2024). What Effects the Generalization in Visual RL: Policy Consistency with Truncated Return Prediction (TRP). [3]
* Grooten, B., et al. (2024). MaDi: Learning to Mask Distractions for Generalization in Visual Deep RL [4]
* Laskin, M., Srinivas, A., & Abbeel, P. (2020). CURL: Contrastive Unsupervised Representations for Reinforcement Learning. ICML. [5]
* Yarats, D., Fergus, R., Lazaric, A., & Pinto, L. (2021). Mastering Visual Continuous Control: Improved Data-Augmented Reinforcement Learning (DrQ-v2) [6]
* Hansen, N., Su, H., & Wang, X. (2021). SVEA: Stabilizing Deep Q-Learning with ConvNets and ViTs under Data Augmentation. NeurIPS. [7]
* Hansen, N., & Wang, X. (DMC-GB) and Almuzairee, A., et al. (DMC-GB2). DMControl Generalization Benchmarks. [8]
* Bertoin, D., Zouitine, A., Zouitine, M., & Rachelson, E. (2022). Look where you look! Saliency-guided Q-networks for generalization in visual RL (SGQN). NeurIPS. [9]
* Chen, C., Xu, J., Liao, W., Ding, H., Zhang, Z., Yu, Y., & Zhao, R. (2024). Focus-Then-Decide: Segmentation-Assisted Reinforcement Learning (FTD). [10]
* Huang, S., Sun, Y., Hu, J., Guo, S., Chen, H., Chang, Y., Sun, L., & Yang, B. (2023). Learning Generalizable Agents via Saliency-Guided Features Decorrelation (SGFD).  [11]
* Tao, S., et al. (2024). ManiSkill3: GPU Parallelized Robotics Simulation and Rendering for Generalizable Embodied AI. arXiv:2410.00425. [12]

**Questions:**

- How does the method compare to RAD, DrQ-v2, SVEA, and CURL under their recommended settings? Please report both ID performance and ID and OOD gaps, for example you can use the Distracting Control Suite.
- If reward spread is small, could you form positives/negatives using value estimates or TRP targets? Any preliminary results?
- Please provide a compute table (SPS and wall-clock) for baseline CNN, +gaze (no contrast), and full method, including buffer maintenance and mining time, across several tasks.

---

> ### Author Response · Authors · 2025-11-21
> **Official Response to Reviewer iVWx**
>
> We thank the reviewer for the detailed assessment and for raising several important points. We appreciate the opportunity to clarify our design choices and expand on the scope of our experiments. Please find our responses below.
>
> 1. Regarding comparison to RAD, DrQ-v2, SVEA, and CURL
>
> Thank you for raising this important point. We would like to clarify that our method targets a different problem class from augmentation-based or invariance-based visual RL methods such as RAD, DrQ-v2, SVEA, and CURL. These approaches are designed to improve representation robustness under pixel-level perturbations through data augmentation, whereas our method explicitly learns spatial attention through a return-guided objective. The two techniques therefore operate on different levels of the visual stack:
>
> -  Augmentation-based methods mainly improve robustness by making the encoder invariant to pixel-level perturbations and appearance changes.
>
> - Our method aims to improve spatial credit assignment by learning where the agent should focus to achieve high returns.
>
> Because of this distinction, these methods are not direct substitutes; rather, they are potentially complementary, and can even be combined in principle (although merging the two methods need care because both introduce auxiliary contrastive objectives).
>
> We fully agree that including stronger pixel-RL baselines helps contextualize our contributions. We are currently running CURL + SAC on a subset of ManiSkill3 manipulation tasks. Preliminary runs indicate that augmentation-heavy methods struggle to stabilize on contact-rich 3D manipulation (consistent with prior observations in Hansen et al. 2021, Yuan et al. 2022), but we will report full curves as soon as the experiments complete within the rebuttal period.
>
> Also, as the reviewer mentioned, evaluating our attention module on DMControl Generalization Benchmarks and Distracting Control Suite is an interesting direction. These benchmarks differ substantially from manipulation as they emphasize camera perturbations and background shifts. We plan to include a set of representative experiments on these domains in the revised version to clarify the complementary strengths of both families of approaches.
>
> We will update the commend as the results become available in the coming days.
>
> 2. Regarding using value estimates or TRP targets when reward spread is small
>
> Value estimates or TRP targets could in principle be used as alternative supervisory signals, but in practice they are not well suited for our attention-learning setting. Early in training, value predictions are often highly unstable, so using them to drive the contrastive loss would inject substantial noise. More fundamentally, in sparse-reward environments the reward stream provides almost no intermediate signal, which means that both value and TRP estimates inevitably inherit this sparsity. Their predictions tend to be flat or dominated by noise and therefore offer little guidance about where attention should focus.
> We note that strictly sparse rewards are inherently misaligned with our return-guided contrastive learning framework, since any target derived from the reward signal (returns, values, or TRP) lacks the variation needed to form informative positive/negative pairs. We acknowledge this as a limitation of the current approach, and exploring alternative supervision sources that do not rely directly on reward magnitude for attention learning represents a promising direction for future work.
>
> 3. Regarding the compute table
>
> We will include the full compute table (SPS, wall-clock time, buffer maintenance overhead, and triplet-mining cost) in the appendix of the revised version.
>
> We hope these clarifications address the reviewer’s concerns and help better position our method relative to existing approaches. We will update the official comment with the additional experiments and comparisons as soon as the ongoing runs complete. Please let us know if further details would be helpful.

---

> > ### Comment · Area_Chair_CEsH · 2025-11-22
> >
> > Hi Reviewer,
> >
> > The authors have submitted their responses to your reviews. Please take a look and let the authors know if you have any further questions or concerns. Thank you again for your contributions to ICLR!
> >
> > Best regards,
> > AC

---

> > ### Author Response · Authors · 2025-11-28
> > **Follow-up Response to Reviewer iVWx**
> >
> > We thank the reviewer again for the insightful suggestions. Your feedback was extremely helpful in clarifying the positioning of our work and sharpening the contributions.
> >
> > As mentioned in our earlier comment, we have now updated our SAC experiments to include a direct comparison against CURL under its recommended settings. The corresponding results and discussions are now included in Section 5.5, and the implementation details for CURL are provided in Appendix C.
> >
> > We would like to reiterate that our framework aims to learn where to focus during reinforcement learning, particularly in manipulation environments where spatial credit assignment is crucial. By contrast, augmentation-based and invariance-driven methods such as RAD, DrQ-v2, SVEA, and CURL primarily strengthen encoder robustness to pixel-level perturbations. These two families operate on different layers of the visual stack and are therefore not direct substitutes; rather, they can be complementary, as they address distinct challenges in visual RL.
> >
> > Regarding the compute table, we now include a full breakdown of the computational cost, including SPS, wall-clock time, foveal forward cost, and contrastive mining overhead in Appendix B.
> >
> > Thank you again for your insightful feedback and constructive suggestions. Your comments greatly helped us clarify the scope of our method, refine the narrative, and strengthen the overall presentation of the paper. Please let us know if there is anything else we can clarify.

---

### Author Response · Authors · 2025-12-03
**Official Comment to Area Chair and Reviewers**

We thank all the reviewers for the thoughtful evaluation and feedback on our original submission. Following the reviewers’ feedback, we have revised the manuscript and addressed all major concerns raised across the four reviews. The major changes are highlighted in blue in the revised manuscript and below is a summary of the response and changes:

**1. Stronger pixel-based baselines and additional comparisons (iVWx, 6vmd, LYqW)**

We agree that this is an important comparison for positioning our method within the broader visual RL augmentation methods.

We have now added:
- CURL + SAC under its recommended settings (Section 5.5, Figure 4)
- CURL implementation details (Appendix C)
- An additional research question (RQ5) addressing how our method compares to contrastive learning baselines (Section 4.1, Section 5.5)

**2. Clarification and improvement of return-guided triplet mining (6vmd, eqoC)**

- Introduced an anchor-centered sampling procedure that explicitly partitions retrieved neighbors into higher- and lower-return groups relative to the anchor, ensuring consistent return ordering and eliminating the ambiguity present in the original median-based split.
- Rewritten Sections 3.3.1-3.3.2 and 4.1–4.2 to provide clear implementation details
- Described how the FAISS index, return buffer, and online triplet construction function (Section 3.3.1-3.3.2)
- Re-evaluated all experiments using the corrected sampling mechanism (Section 5)

The updated results show reduced variance and improved consistency across tasks, showing that the revised method addresses the ambiguity highlighted by reviewers.

**3. Sparse reward limitation & assumption clarity (iVWx, LYqW)**

Our return-guided contrastive objective relies on some variation in returns to form informative positive and negative pairs, therefore in strictly sparse-reward settings where most trajectories receive identical or near-identical returns, naturally weaken the contrastive signal. We have clarified this limitation in Section 3.3.2 and added a detailed discussion in Section 6.

Importantly, this assumption is consistent with common practice in manipulation RL, where shaped or semi-dense reward structures provide meaningful return diversity due to multi-stage task progress and partial-success signals. The revision makes the intended scope of our method clearer while also contextualizing the limitation which is inherent to return-based contrastive learning in general rather than specific to our formulation.

**4. Multi-object / multi-fovea behavior (LYqW, eqoC)**

- Multi-object behavior: We explained that in multi-object tasks, the model naturally shifts its foveal focus across relevant objects as the episode progresses, driven by the multi-stage structure of shaped manipulation rewards.
- Extension to multi-fovea: We added a discussion in Section 6 describing how multiple foveal regions (e.g., a mixture of Gaussians or multiple attention heads) could be incorporated in future work to handle scenarios requiring simultaneous focus on multiple spatial regions.

**5. Additional implementation details and ablations**

- The “Foveal attention (no contrastive)” ablation is now more clearly described (Section 5.1)
- We expanded details regarding buffer maintenance, FAISS updates, and return storage (Section 3.3.2)


**6. Compute overhead and wall-clock performance (iVWx)**

We included a full compute breakdown (SPS, wall-clock, attention cost, FAISS cost) in Appendix B.

**7. Positioning within visual RL**

To address conceptual clarifications requested across multiple reviewers, we have strengthened the positioning of our method within the broader visual RL methods:
- Pixel-invariance vs. spatial credit assignment: We now clearly distinguish our approach from augmentation-based methods such as RAD, DrQ-v2, SVEA, and CURL, which primarily enforce pixel-level invariance through random crops or perturbations.
- Our return-guided spatial attention: We emphasize that our framework instead focuses on spatial credit assignment, using return-guided attention to highlight task-relevant regions rather than encouraging invariance to them.
- Revised related works (Section 2) and Section 5.5: These sections have been rewritten to more explicitly explain this conceptual difference and to place our method as complementary to, rather than overlapping with, invariance-based visual RL approaches.


The revised manuscript now incorporates extensive clarifications, stronger baselines, refined methodology, and a detailed discussion of limitations and scope. We believe these revisions substantially improve the clarity, positioning, and empirical grounding of the paper. We thank all the reviewers for their feedback, which helped us significantly strengthen the work.

---

### Meta-Review · Area_Chair_Ahwv · 2026-01-07

**Summary:**

Reviewers had a number of major concerns about the paper ranging from a lack of adequete comparisons to visual representation learning in reinforcement learning, a lack of evaluation of OOD generalization of the approach, a lack of adequete discussion of related work, and the scarcity of the required triple data needed to train the approach. The authors were able to address some of the concerns, such as adding an additional baseline to the visual representation in reinforcement learning. However, the remaining set of concerns in the paper makes it so that the paper is not ready for publication in the current state.

**Reviewer Concerns:**

Reviewers had a number of major concerns about the paper ranging from a lack of adequete comparisons to visual representation learning in reinforcement learning, a lack of evaluation of OOD generalization of the approach, a lack of adequate discussion of related work, and the scarcity of the required triple data needed to train the approach, as well as various textual clarifications.

Reviewers were able to partially address some of the concerns on textual clarifications as well as adding a CURL baseline. However, a more extensive comparison to other visual representation learning methods in RL is missing, as well as a revised discussion about these baselines in the related work. Finally, concerns about the evaluation of the OOD generalization of approach is not addressed.

**Reviewer Scores:**

I believe that the concerns of Reviewer iVWx was not adequetly addressed, with no additional evaluations of OOD generalization, and believe the score would remain a reject. In addition, the major concern of Reviewer 6vmd on comparisons with other relevant baselines is also not adequetly addressed, with only one additional set of results using the CURL method. Finally, a major concern of Reviewer  eqoC
 on the scarcity of triple data is not well addressed, with benchmarks such as RL-Bench and Mani-skill not having dense reward data that could be used to learn this foveal attention layer.

---

### Decision · Program_Chairs · 2026-01-26

Reject